# Non-Thermal Ammonia Decomposition for Hydrogen Production over Carbon Films under Low-Temperature Plasma—In-Situ FTIR Studies

**DOI:** 10.3390/ijms23179638

**Published:** 2022-08-25

**Authors:** Julia Moszczyńska, Xinying Liu, Marek Wiśniewski

**Affiliations:** 1Physicochemistry of Carbon Materials Research Group, Faculty of Chemistry, Nicolaus Copernicus University in Toruń, Gagarina 7, 87-100 Torun, Poland; 2Institute for Development of Energy for African Sustainability, University of South Africa, Private Bag X6, Florida 1710, South Africa

**Keywords:** carbon film, non-thermal plasma, ammonia splitting, catalysis

## Abstract

Due to easy storage and transportation, liquid hydrogen carriers will play a significant role in diversifying the energy supply pathways by transporting hydrogen on a large scale. Thus, in this study, amorphous carbonaceous materials have been employed for hydrogen production via ammonia decomposition under non-thermal plasma (NTP) conditions. The adsorption and splitting of ammonia over carbons differing in the chemical structure of surface functional groups have been investigated by in situ spectral studies directly under NTP conditions. As a result of NH_3_ physical and chemical sorption, surface species in the form of ammonium salts, amide and imide structures decompose immediately after switching on the plasma environment, and new functionalities are formed. Carbon catalysts are very active for NH_3_ splitting. The determined selectivity to H_2_ is close to 100% on N-doped carbon material. The data obtained indicate that the tested materials possess excellent catalytic ability for economical, CO_x_-free hydrogen production from NH_3_ at a low temperature.

## 1. Introduction

The growth of technological development has caused a huge increase in energy demands. People have used fossil fuels such as coal, natural gas, or petroleum for ages. Now we know that burning them contributes to environmental degradation through the emission of greenhouse gases, especially CO_2_ [1]. The increase in ecological awareness has caused a constantly growing need for alternative and renewable energy resources [2]. Solar panels and windmills are extremely favorable from the environmental point of view, but they need specified conditions to achieve good efficiency. Because of that, the question arises of how to obtain green energy to satisfy our needs. The answer is hydrogen, especially green hydrogen [3].

Nowadays, hydrogen production is no longer a problem; it is possible with water electrolysis. This was explored by Liu et al. [4] who investigated kinetics and oxidation pathways using Fe^3+^ as a catalyst. Literature data show that water electrolysis is the most preferred method of producing green hydrogen [5,6]. However, the main problem of hydrogen use is its distribution, which requires a lot of energy. There are a few different ways to store and transport hydrogen. One way is by compressing hydrogen gas, but it is energy inefficient. The second way is a carriage of liquefied H_2_, but it requires an extremely low temperature −250 °C. Recently, significant attention has been paid to obtaining hydrogen from ammonia decomposition, the third method to store H_2_. It is the most advantageous way, as mentioned above.

Pinzón et al. [7] obtained CO_x_-free hydrogen by catalytic ammonia decomposition using Ru supported on reduced graphene oxide (rGO). Ruthenium is often used as a catalyst; nevertheless, its cost makes the process unprofitable. Bell et al. [8] developed a cheaper catalyst which was cobalt deposited on γ-Al_2_O_3_. They investigated the dependence between cobalt particle size and catalytic activity.

One of the most commonly used catalysts utilized in the ammonia decomposition are iron and cobalt (e.g., [9]). Unfortunately these metals tend to sinter thus need a promoter to maintain high activity. In this case, the authors [9] used lanthanum oxides.

To our knowledge, only a few studies refer to ammonia decomposition by non-thermal plasma (NTP). Lu et al. [10] investigated the influence of dielectric barrier discharge plasma on the hidden active phase. They compared thermal catalytic ammonia decomposition and splitting in plasma. El-Shafie et al. [11] obtained hydrogen by using plasma in a plasma membrane reactor. The authors applied a palladium-copper membrane to separate hydrogen. This work focused on the research of energy and exergy of this process. El-Shafie et al. [12] explored alumina particles as a catalyst in the ammonia decomposition process by a dielectric barrier discharge reactor. Akiyama et al. [13] obtained pure hydrogen without by-products by using non-thermal, atmospheric-pressure plasma. They applied different types of electrodes.

Bearing in mind (i) high stability under the H_2_ atmosphere, (ii) high surface area, (iii) well-defined pore structure, (iv) moderate N-affinity, we decided to look closer at the possibility of using the carbonaceous materials for NH_3_ splitting under the NTP environment.

The present study aimed to prove the catalytic ability of carbon materials. In this work, for the first time, the splitting of ammonia over carbons differing in the chemical structure of surface functional groups has been investigated by in situ FTIR spectroscopy. As a result of carbon surface exposition to the NH_3_ atmosphere, the physically and chemically adsorbed ammonia molecules are detected on the surface of carbon films. The evolution of these functionalities under NTP conditions is observed.

## 2. Results

### 2.1. Carbon Films Preparation

The infrared spectrum of C_des_—carbon sample outgassed at 800 °C (Figure 1a) shows the presence of two main groups of signals sharp at ca. 1600 1/cm and mutually overlapping in the range 1100–1500 1/cm. These signals are due to C=C and C–C stretching modes that are weakly active in the IR because of the breakdown of selection rules [14]. Moreover, some C–H stretching vibrations can be detected at 3059 1/cm—a characteristic region of unsaturated and aromatic units. Note that there are no aliphatic functionalities in the carbon structure.

Doping nitrogen (5.31 m/m% based on elemental analysis) to the material (CdesN) does not drastically change the carbon FTIR spectrum (Figure 1b). Only the decrease in the background level and a shift of continuous absorption limit in the direction of shorter waves is observed due to the electrodonating character of newly formed surface species.

Further blue-shift of continuous absorption limit is observed for the carbon sample after oxygen doping. Additionally, the presence of absorption bands at 1830, 1760, and 1200, 1/cm for this sample (Figure 1c) indicates that some acidic surface groups are cyclic anhydrides. These surface structures were discussed previously, e.g., [15,16,17,18].

The spectral changes caused by ammonia adsorption (Figure 1a’–c’) on these materials are negligible; thus they are much more visible as the differential spectra and will be discussed in the following sections.

### 2.2. In-Situ Plasma-Assisted Catalytic NH_3_ Conversion Investigations

The FTIR spectra of the gaseous products formed as a result of the NH_3_ decomposition over tested carbon films (30 min in static conditions) at 1 kW/m are shown in Figure 2. The NH_3_ conversion, as was expected, is NTP power-dependent. The higher power, the higher conversion close to 100% was reached for both high temperature-treated carbons (C_des_ and C_des_N). The differences between the carbons were observed for these samples under lower NTP power. The highest catalytic activity was determined for the nitrogen-doped sample (C_des_N). It is important to note that this material is characterized by the highest H_2_ selectivity (over 98%). Only ca. 40 ppm of CH_4_ and no C_2_H_2_ were detected after 30 min reaction under the most extreme conditions.

The presence of surface oxygen functionalities causes the lower activity and selectivity of oxidized material (C_ox_). It will be discussed in the following paragraphs.

Bearing in mind the high resistance of carbons to the reducing atmosphere (including H_2_) it wasn’t a surprise that both treated at 600 °C catalysts (C_des_ and C_des_N) show no structural changes caused via NH_3_ splitting reaction. Raman spectra, as well as high-resolution SEM pictures, remain unchanged, proving no effect of the process on sample morphology (Figure 3).

In contrast, for the oxidized material (C_ox_), the intensity of the D-band slightly rises as the G-band decreases in intensity. Also, the SEM analysis reveals that the reaction somewhat destroyed the surface. Thus it seems reasonable to look closer at the overall mechanism.

While ammonia is adsorbed on carbonaceous material, some characteristic surface species are formed. As a result of NH_3_ adsorption (Figure 4, red spectrum), broad, mutually overlapped IR bands in the region of OH and NH stretching vibrations are formed at 3456, 3381, 3207, 3031 1/cm. The shape (wideness) of these bands and the presence of a signal at 2800 1/cm indicate the formation of hydrogen bonds. Their origin may be both: (i) hydrogen bonds between adsorbed NH_3_ molecules and oxygen-functional groups of C_ox_, or (ii) adsorbate–adsorbate-type interactions between physically adsorbed NH_3_ and previously chemisorbed molecules.

The spectral changes in the region of C=O stretching vibrations are caused by ammonia chemisorption on the oxidized carbon surface. The disappearance of band doublet at 1841 and 1753 1/cm is due to the reaction between NH_3_ and surface anhydrides. One of the products thus formed is surface ammonium salts of carboxylic acids. In their IR spectrum, an intense band at 1471 1/cm (δ_s_ of NH_4_^+^ ion) and bands at 1551 1/cm (ν_as_ of COO^−^), and 1397 1/cm (ν_s_ of COO^−^) are formed. Such observations suggest that adsorption on Brönsted acid sites becomes dominant on this surface. However, the presence of the signal at 1655 and ca. 1300 1/cm (visible as a right shoulder of 1397 1/cm band) suggests that some Lewis sites are also present on the C_ox_ surface. The IR band at 1107 1/cm can be attributed to physically adsorbed NH_3_.

When NTP is turned on, the IR bands of structures mentioned above disappear immediately, indicating that the NH_3_ decomposition occurs effectively.

In the NH stretching region, the IR peaks at 3456 and 3381 1/cm of –NH_2_ structures remain stable. A new signal at 3528 1/cm appears. In the fingerprint spectral region, the intensity of the negative doublet bands decreases, meaning that surface anhydrides are at least partially rebuilt. Interestingly, new IR signals appear at 1695 and 1220 1/cm. Such observations indicate that surface amines (C-NH_2_) structures are formed.

A somewhat different IR spectrum was registered when NH_3_ was adsorbed on outgassed at 600 °C carbon film (Figure 5—red spectrum). In this case, besides the physical adsorption (1096 1/cm), the interactions with Lewis acid centers dominate (1588, 1260 1/cm) combined with strong hydrogen bonds—a broad band in the range of 3300–2400 1/cm.

Very interesting seems to be the presence of the IR bands below 900 1/cm. While the signal at 867 1/cm can be attributed to NH_3_^δ–^ the others at 661 and below 370 1/cm prove the formation of the secondary and tertiary amines, due to dissociative adsorption of NH_3_ on a positively charged surface.

After switching on the NTP, the physically adsorbed NH_3_ molecules (1096 1/cm) decompose while the bands attributed to adsorption on Lewis sites (1553 and 1201 1/cm) increase in intensity. Similarly, the 661 and 425 1/cm bands rise in intensity, too. Note that the signal attributed to C=C stretching mode at 1600 1/cm decreased in intensity.

Similar spectral changes confirming dissociative NH_3_ adsorption and splitting under NTP conditions were observed during the reaction over C_des_N surface (Figure 6). Some differences arising from the other surface composition caused by N-doping were observed in the fingerprint region—below 900 1/cm.

## 3. Discussion

Due to easy storage and transportation, liquid hydrogen carriers will play a significant role in diversifying the energy supply corridor by transporting hydrogen on a large scale. Low or even no-carbon footprint is the crucial factor for their application. Of all the liquid hydrogen carriers, ammonia has proven to be a carbon-free and high-energy density sustainable candidate. The NH_3_ splitting has the advantages of one-way transport. Produced in this way, N_2_ does not need to be directly recovered and recycled after the dehydrogenation step.

In this work, we proved that ammonia cracking might generate environment-friendly nitrogen and hydrogen not only on the metal-based catalysts, as was stated previously by Wei et al. [19] and the group of Ohtsuka [20]. It has been reported that the Fe-catalyzed NH_3_ decomposition may proceed through a mechanism involving the N-containing intermediate species. We showed here that a similar mechanism is also possible on carbon catalysts but contrary to metallic systems reaction leads here on the surface maintaining the carbon structure (Figure 3, Figure 4, Figure 5 and Figure 6).

We proved that effective NH_3_ splitting on the carbon surface was observed on both high-temperature treated materials. This reaction was examined in the literature over Fe and Co containing active sites [10,21,22]. However no work was done for pure carbons, nor for NTP conditions.

Outgassed carbon surfaces (C_des_ and C_des_N) played the role of nitrogen scavengers. As a consequence, functionalities similar to carbon nitrides were formed. Interestingly, doping oxygen hampered the overall process, most probably via competition of O and N connected to the active center. Thus, a bare or N-containing carbon surface seemed ideal for the dissociative adsorption of NH_3_ under NPT conditions. Once the NH_3_ decomposed, the isolated N or H adsorption became important. The properties of the carbon surface made the interaction of N much stronger than that of H with C. Nitrogen atoms tended to occupy the active sites by preference, which could lead to catalytic deactivation. Nevertheless, such an effect was not observed in this study; the selectivity on the C_des_N catalyst was as high as 98%. However, further examination of the circulation of N “on” and “in” the C-surface needs to be carried out.

## 4. Materials and Methods

### 4.1. Carbon Films Synthesis

Carbon films utilized in this study were prepared from cellulose. The tested materials shows the 2-dimensional nature of, i.e., unlimited length and width (in this study, 35 × 15 mm) and a thickness of about 20 µm (Figure 1). The charring experiments were set up as follows: cellulose films were carbonized at 600 °C for 1 h under a dynamic vacuum. These samples were labeled—C_des_. To prepare an oxidized carbon film (C_ox_), the C_des_ samples were exposed to 100 kPa of O_2_ at 300 °C for 1 h and then evacuated at 200 °C. The nitrogen-doped carbons (C_des_N) were prepared from C_ox_ materials by annealing in an NH_3_ atmosphere (100 kPa) at 600 °C for 1 h. For catalytic tests, the powdered carbon samples were prepared similarly.

### 4.2. Materials Characterization

The full characteristics of carbon films, including elemental and thermogravimetric analysis, low-temperature N_2_ adsorption, ^13^C NMR, and structure modeling, were presented recently [23,24,25]. The results indicated that carbon films are microporous solids with a bimodal pore size distribution. The pores are grouped around two main diameters. The first diameter approaches 0.5–0.6 nm, and the second diameter occurs in the range 1.3–1.5 nm. Thus, the studied carbon films possess interesting features, i.e., nearly homogeneous microporosity, that gives the carbon films a molecular sieve character.

The surface area and structure of carbon films contribute to the catalytic activity only if they exert some control on the accessibility of reactants towards the active centers. There is no enormous difference between the pore size distribution of doped and non-oxidized carbon samples [26]. Thus we could treat the carbon films as a microporous material with similar porosity and focus more on the surface chemistry instead of micropore structure.

Carbons were also characterized with scanning electron microscopy (SEM), using a Quanta 3D FEG (EHT = 30 kV) instrument and Raman spectroscopy. The spectra were measured with the Senterra micro-Raman (Bruker Optiks, Billerica, MA, USA) spectrometers. The spectral parameters were as follows: laser power, 2 mW at 532 nm; resolution, 4 cm^−1^; CCD temperature, –64 °C; laser spot, 2 μm; and total integration time, 100 s (50 × 2 s); objective 20× was used.

### 4.3. In-Situ Plasma-Assisted Catalytic NH_3_ Conversion Investigations

The IR spectroscopic studies were carried out in a vacuum cell described previously [20,26,27,28], and plugged into the vacuum line (Figure 7). For the in situ IR study, ammonia (constant pressure 1.38 Torr) was exposed to a carbon film at 1 kW/m normalized power NTP atmosphere. The spectra were collected during a 30 min reaction. Typically, 32 scans were collected with an instrument resolution 4 1/cm. The IR cell was made of quartz with KBr windows. The cell construction enabled the thermal treatment of the carbon film up to 1000 °C in any controlled atmosphere or in a vacuum.

The composition of the gas phase in contact with the catalysts was also monitored by Mattson Genesis II FTIR spectrophotometer. A carbon film sample in the IR cell could be moved in and out of the infrared beam. Single beam spectra were recorded with the carbon film sample either in the path of the infrared beam or withdrawn from the beam, as well as with and without the gases in the cell. Subtracting the spectrum of the gas phase (recorded within 15 s) from that of the combined gas and surface species gives the spectrum due to surface species adsorbed on carbon samples.

The other important element of the system is the glow discharge (GD) reactor operating under 10 kV (30 mA) and 30 kHz conditions. The distance between electrodes was 30 cm. The effluent gases (NH_3_ and C_x_H_y_) were analyzed quantitatively using a multiple-pass gas cell (with the pathlength of 2 m) attached online to the flow system (Figure 7). The selectivity was calculated from relation H_2_/(H_2_ + ΣC_x_H_y_). Note that no carbon oxides were detected within all the experiments.

## 5. Conclusions

In this work, for the first time, we examined a promising method of obtaining pure hydrogen by ammonia decomposition in a non-thermal plasma environment over carbon catalysts. We showed the direct experimental evidence that carbon works as an N scavenger, but the bonding was not so efficient due to the low electronegativity difference.

The process was performed only at the surface; we did not interfere with the C-structure. Based on the presented results, one could expect that the used catalysts were extremely stable under NTP conditions, although further studies are needed.

Carbons can be harnessed effectively as catalysts for hydrogen production. The data obtained indicate that the tested materials possess excellent catalytic ability for economical, CO_x_-free hydrogen production from NH_3_ decomposition at a low temperature.

## Figures and Tables

**Figure 1 ijms-23-09638-f001:**
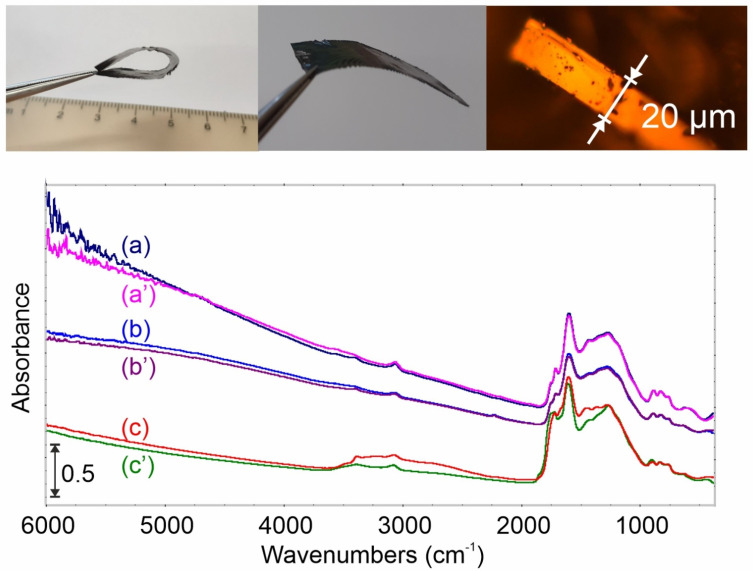
Upper panel—images of carbon film; bottom panel—FTIR spectra of tested materials (a) C_des_, (b) C_des_N, (c) C_ox_, respectively a, b, c, before, and a’, b’, c’ after NH_3_ adsorption.

**Figure 2 ijms-23-09638-f002:**
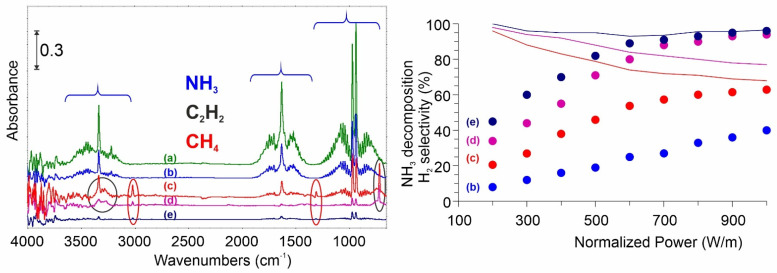
Left panel: FTIR spectra of the gas phase before (a) and after the NH_3_ splitting process over tested carbons: (b) no catalyst, (c) C_ox_, (d) C_des_, (e) C_des_N. Right panel: catalytic conversion and selectivity of tested systems.

**Figure 3 ijms-23-09638-f003:**
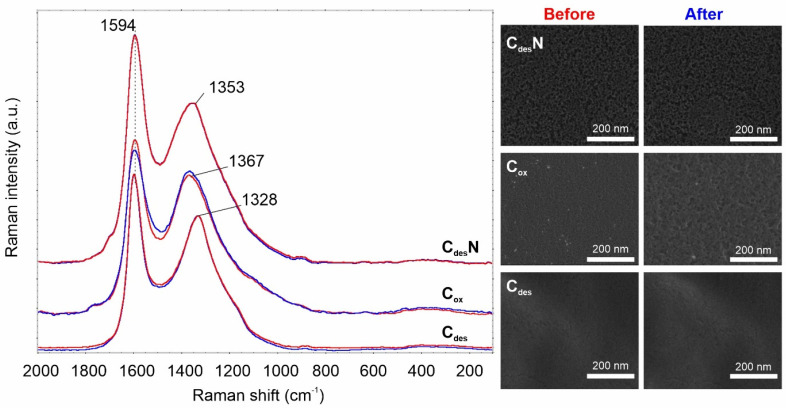
Raman spectra and SEM pictures of tested materials before (red) and after (blue) 30 min reaction under NH_3_ NTP.

**Figure 4 ijms-23-09638-f004:**
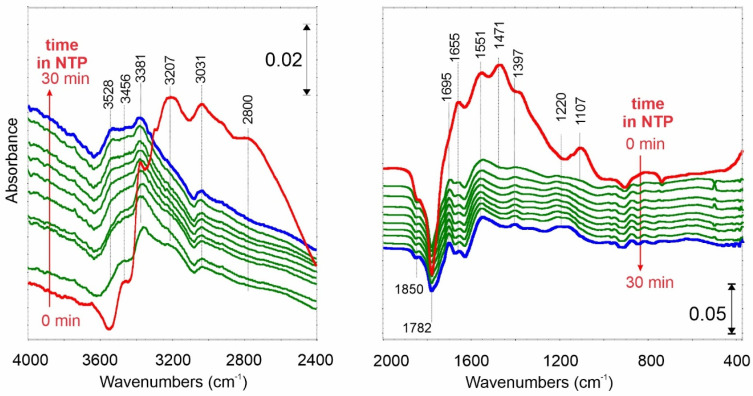
FTIR differential spectrum of C_ox_ carbon film after NH_3_ adsorption at 25 °C (red spectrum—0 min) and time-resolved spectra when NTP is switched on from 0 to 30 min.

**Figure 5 ijms-23-09638-f005:**
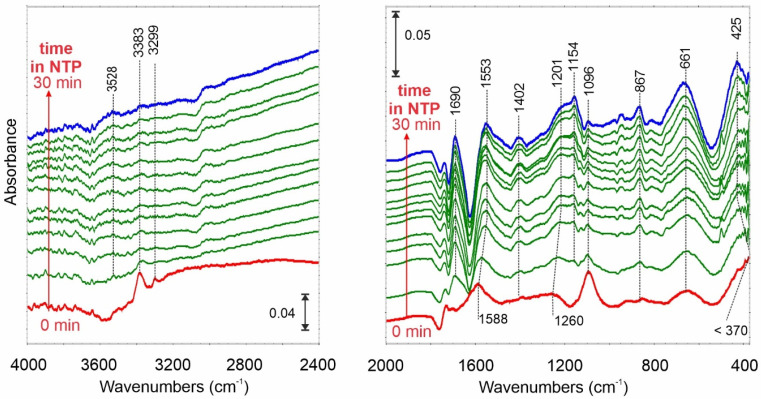
FTIR differential spectrum of C_des_ carbon film after NH_3_ adsorption at 25 °C (red spectrum—0 min) and time-resolved spectra when NTP is switched on from 0 to 30 min.

**Figure 6 ijms-23-09638-f006:**
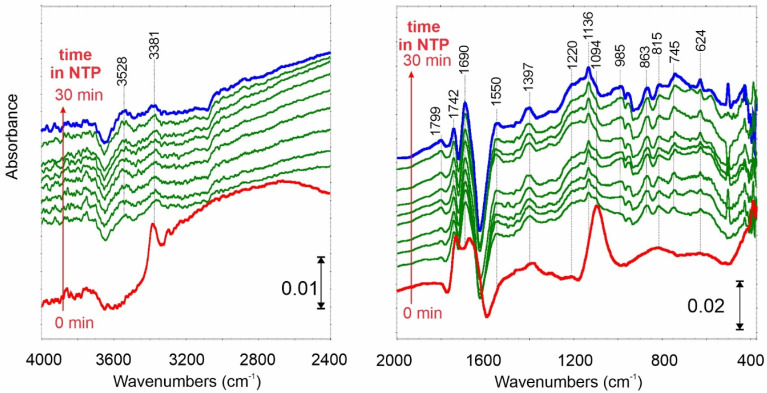
FTIR differential spectrum of C_des_N carbon film after NH_3_ adsorption at 25 °C (red spectrum—0 min) and time-resolved spectra when NTP is switched on from 0 to 30 min.

**Figure 7 ijms-23-09638-f007:**
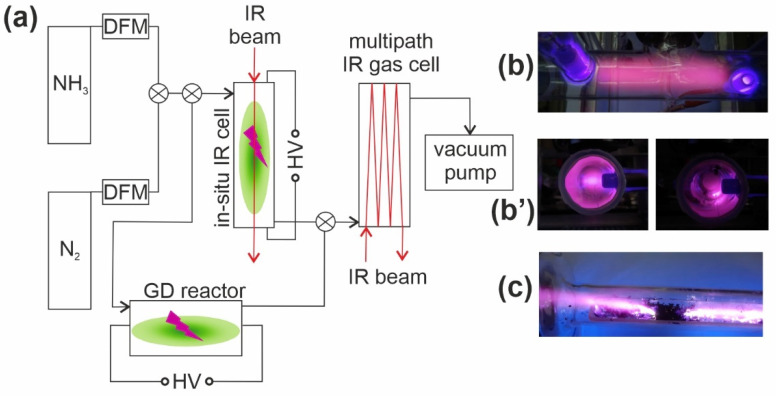
(**a**) scheme of an in situ plasma-assisted catalytic NH_3_ conversion system with pictures of working under NTP conditions IR cell (**b**), (note that pictures in (**b’**) were taken in parallel to IR beam) and glow discharge (GD) reactor (**c**).

## Data Availability

Data presented in this study are available from the corresponding author on reasonable request.

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
