# Peer review of "Non-Thermal Ammonia Decomposition for Hydrogen Production over Carbon Films under Low-Temperature Plasma—In-Situ FTIR Studies"

_ijms, 2022, doi:10.3390/ijms23179638_

Round 1

Reviewer 1 Report

The article describes the research on non-thermal ammonia decomposition for hydrogen production over carbon films under low-temperature plasma - in-situ FTIR studies. In many places, the article contains inaccuracies that need to be corrected. The article is not suitable for publication in IJMS as it stands.

General remarks

1. The introduction contains many generalities. Please rewrite this chapter and make more specific, not slogans that do not explain much.

2. What are the advantages and disadvantages of the presented methods of obtaining hydrogen?

3. A separate subchapter should clearly define the purpose and scope of the research.

4. Why do the Authors go straight to the results in Chapter 2, if they did not describe the methodology and research scheme?

5. The Materials and Methods section should appear before the Results section.

6. Conclusions should be bulleted and contain specific research findings.

The article is not well written and needs improvement.

Specific remarks

7. 16 line: Please explain FTIR abbreviation in the article.

8. 30 line: Please indicate why the listed energy sources are ineffective under certain conditions.

9. 32 line: Why is there a question mark at the end of the sentence and what the authors mean by "green energy".

10. 33 line: What is the difference between hydrogen and "green hydrogen". Please explain in the article.

11. 43 line: What is the difference in energy expenditure between methods of obtaining hydrogen? What exactly are the differences between these methods of hydrogen production?

12. 45 line: Use the full names mentioned for the first time in the article.

13. 46 line: Explain the difference in costs.

114. 51 line: It is unclear whether the studies by Xun et al. or in the research of the authors of the article.

15. 55 line: What was the specific efficiency of the heating installation ?

16. 56 line: What's the price difference between green ammonia and regular ammonia ?

17. 62 line: Explain PMR acronym.

18. 64 line: Please explain DBD abbreviation in the article.

19. 66 line: What was the energy efficiency of this method ?

20. 68 line: In which specific article ?

21. 72 line: What does this sentence mean. Completely incomprehensible.

22. 77 line: Enter units for 1600.

23. 94 line: What exactly do the top photos show? On the ordinate, what are the Absorbance values?

24. 107 line: On the ordinate, what are the absorbance values ?

25.123 line: No value in the ordinate axis.

26. 144, 166, 181 lines: On the ordinate, what are the values ?

27. 184-188 lines: The authors cite [20] and [21] without mentioning the authors of these works. What is the evidence the authors write about. Please explain specifically in the article.

28. 215 line: Explain abbreviation.

29. 233 line: Fig. 7: What is shown in figure b' ?

30. 252 line: What does Fig. 2 show and where is it ?

I recommend an in-depth review of the manuscript, including comments, to make it an article suitable for publication in the IJMS.

Author Response

attached file

Reviewer 2 Report

The ammonia decomposition method of hydrogen production on the carbon surface is novel and promising. The presented experiment is discussed in detail and convincing.

In the Introduction (p. 1, lines 38-44), the authors mention the storage of hydrogen gas as a problem. I note that the procedure just described does not solve this either.

Question: Is the formation of N-O excluded in the process?

Please check the units:

% m/m --> m/m% ?

kW/m - power/unit what?

I recommend accepting the manuscript for publication.

Author Response

attached file

Reviewer 3 Report

Only the process of decomposition NH3 has been studied. Where is production H2? What is the aim of the study?

Author Response

attached file

Round 2

Reviewer 3 Report

Needs a little improvement.

There is one big mistake. Carbon is an element, not material. The phrase must be completed

Author Response

Dear referee, you are right only in a half. "Carbon", indeed is an element, but it is not a mistake to use "carbon" for carbonaceous materials. Please see, "Carbon" (journal).
Similarly, one could accuse using "metal oxide" for materials on which  -OH surface functionalities are present, are (almost) unremovable, and are of decisive importance for its properties.